# Dysfunction of Complementarity Determining Region 1 Encoded by T Cell Receptor Beta Variable Gene Is Potentially Associated with African Swine Fever Virus Infection in Pigs

**DOI:** 10.3390/microorganisms12061113

**Published:** 2024-05-30

**Authors:** Jiayu Li, Huiyan Xing, Kai Liu, Ninglin Fan, Kaixiang Xu, Heng Zhao, Deling Jiao, Taiyun Wei, Wenjie Cheng, Jianxiong Guo, Xiong Zhang, Feiyan Zhu, Zhigao Bu, Dongming Zhao, Wen Wang, Hong-Jiang Wei

**Affiliations:** 1Key Laboratory for Porcine Gene Editing and Xenotransplantation in Yunnan Province, Yunnan Agricultural University, Kunming 650201, China; hnnydxsyljy@126.com (J.L.); xinghuiyan5433@163.com (H.X.); 18314383380@163.com (K.L.); funnyl571271512@163.com (N.F.); tsljmuch@163.com (K.X.); hengzhao2014@126.com (H.Z.); jiaodeling@163.com (D.J.); weitaiyun@126.com (T.W.); cwj210365@163.com (W.C.); 18788541129@163.com (J.G.); zhangxiong202404@163.com (X.Z.); yunnanfeiyan@163.com (F.Z.); 2Xenotransplantation Engineering Research Center in Yunnan Province, Yunnan Agricultural University, Kunming 650201, China; 3College of Veterinary Medicine, Yunnan Agricultural University, Kunming 650201, China; 4College of Animal Science and Technology, Yunnan Agricultural University, Kunming 650201, China; 5State Key Laboratory for Animal Disease Control and Prevention, National High Containment Facilities for Animal Diseases Control and Prevention, Harbin Veterinary Research Institute, Chinese Academy of Agricultural Sciences, Harbin 150069, China; buzhigao@caas.cn (Z.B.); zhaodongming@caas.cn (D.Z.); 6State Key Laboratory of Genetic Resources and Evolution, Kunming Institute of Zoology, Chinese Academy of Sciences, Kunming 650223, China

**Keywords:** pig, African swine fever virus, *TRBV27* gene, complementarity determining region 1, gene editing

## Abstract

The beta T-cell receptor (*TRB*) expressed by beta T cells is essential for foreign antigen recognition. The *TRB* locus contains a *TRBV* family that encodes three complementarity determining regions (CDRs). CDR1 is associated with antigen recognition and interactions with MHC molecules. In contrast to domestic pigs, African suids lack a 284-bp segment spanning exons 1 and 2 of the *TRBV27* gene that contains a sequence encoding CDR1. In this study, we used the African swine fever virus (ASFV) as an example to investigate the effect of deleting the *TRBV27*-encoded CDR1 on the resistance of domestic pigs to exotic pathogens. We first successfully generated *TRBV27*-edited fibroblasts with disruption of the CDR1 sequence using CRISPR/Cas9 technology and used them as donor cells to generate gene-edited pigs via somatic cell nuclear transfer. The *TRBV*-edited and wild-type pigs were selected for synchronous ASFV infection. White blood cells were significantly reduced in the genetically modified pigs before ASFV infection. The genetically modified and wild-type pigs were susceptible to ASFV and exhibited typical fevers (>40 °C). However, the *TRBV27*-edited pigs had a higher viral load than the wild-type pigs. Consistent with this, the gene-edited pigs showed more clinical signs than the wild-type pigs. In addition, both groups of pigs died within 10 days and showed similar severe lesions in organs and tissues. Future studies using lower virulence ASFV isolates are needed to determine the relationship between the *TRBV27* gene and ASFV infection in pigs over a relatively long period.

## 1. Introduction

African swine fever (ASF) is an acute infectious and hemorrhagic disease that is responsible for a massive number of deaths in pigs and is caused by the African swine fever virus (ASFV) of the genus Asfivirus [1,2]. ASF has resulted in significant economic losses for the pig industry in major pork-trading countries [3]. It is estimated that ASF caused a loss of $267 million to the Russian pig industry in 2011 and $961 million in several European countries [3,4]. As ASF has become a major obstacle for the pig industry worldwide [5,6], it is listed as a notifiable infectious disease by the World Organization for Animal Health (WOAH) (https://www.woah.org/en/disease/african-swine-fever/, accessed on 15 April 2024) [7].

Understanding the mechanisms of pathogen–host interactions is crucial for developing effective vaccines for ASF [8]. Currently, there are no commercially available antiviral drugs or vaccines available to control ASF [9]. To address this situation, many studies have attempted to identify the critical elements closely related to the host cell response to ASFV [10]. ASFV exhibits cellular tropism and replicates primarily in macrophages, which act as antigen-presenting cells (APCs) [11,12]. The entry of ASFV into macrophages is closely associated with macrophage membrane receptors [13]. Previous studies have shown that CD163 is a potential receptor for ASFV, as the virus only infects CD163-positive monocytes [14,15]. However, CD163 knockout pigs do not exhibit the expected resistance to ASFV [16]. In addition, other macrophage membrane receptors, such as MHC II, CD203a, and CD45 were not found to be critical for ASFV infection. This suggests that a comprehensive understanding of the host immune response to virus invasion is urgently needed to combat ASF using immunological approaches [9].

T cells play a crucial role in adaptive immunity and are essential for the clearance of exogenous pathogens [17]. The host immune response is initiated by antigen-presenting cells (APCs), which present pathogen antigens to a heterodimeric T-cell receptor (*TR*) via major histocompatibility complexes (MHCs) [9]. T cells are classified into αβ and γδ types, and the former express TR α (*TRA*) and TR β (*TRB*) chains [18]. The *TRA* and *TRB* chains each contain a variable and a constant domain. Both the variable (V) domains of these two chains have three complementarity-determining regions (CDRs) that are responsible for antigen recognition (Figure 1A). Among them, CDR1 and CDR2 are encoded by germline V genes and interact with the outer α-helices of MHCs [19,20].

*TRB* is characterized by the presence of a multigene family T cell receptor beta variable (*TRBV*). A gene family usually originates from the expansion of an ancestral gene during long-term evolution. The members typically share similar sequence identities and perform comparable biological functions [21,22]. A previous study identified 38 *TRBV* genes in the genome of domestic pigs, all of which encoded three conserved CDRs associated with antigen recognition. In addition, these genes are organized similarly to those of other mammals, despite differences in number and nucleotide polymorphisms [23]. Further comparative genomic analysis revealed that there is variation in the TRBV27 gene between two African suids (*Phacochoerus africanus* and *Potamochoerus porcus*) and domestic pigs. In the latter, the *TRBV27* gene encodes a complete protein that includes CDR1-3 domains. However, the African suids lack a contiguous 284-bp sequence between exons one and two of the *TRBV27* gene, which encodes CDR1 (Figure 1B). This difference has been suggested to be the result of long-term evolution in African suids [24]. However, the correlation between *TRBV27* mutation and changes in the host response to pathogen infection has not yet been established due to a lack of experimental evidence.

In the present study, we hypothesized that the retention of the *TRBV27*-encoded CDR1 region in the domestic pig genome is associated with T-cell immune responses to foreign pathogens. To test this hypothesis, we generated *TRBV27*-edited pigs lacking the sequence covering the CDR1 region using our well-established experimental pig model construction platform [25,26,27,28,29]. The effect of CDR1 deletion on the host response to pathogens was investigated using ASFV as an example.

## 2. Materials and Methods

### 2.1. Ethical Statement

The pigs used in this study were obtained from the Xenotransplantation Engineering Research Center in Yunnan Province. The research protocol was reviewed and approved by the Animal Ethics Committee of Yunnan Agricultural University and the Harbin Veterinary Research Institute (HVRI) of the Chinese Academy of Agricultural Sciences (CAAS) (No. 230223-01-GJ).

### 2.2. Construction of TRBV-Edited Pigs

The mutation of the *TRBV27* gene in African suids was simulated in domestic pigs using the CRISPR/Cas9 system [30,31]. To achieve this goal, four guide RNAs (gRNAs) were designed to target the sequence between exons 1 and 2 of the *TRBV27* gene and inserted into the Cas9 vector (pSpCas9(BB)-2A-GFP, PX458) provided by Professor Xingxu Huang at the University of Shanghai for Science and Technology. To detect the editing efficiency of the sgRNAs, the recombinant vectors were transfected into porcine iliac artery endothelial cells (PIECs) using Lipofectamine 3000 (Thermo Fisher Scientific, Altrincham, UK). The working sgRNAs were then inserted into another Cas9 vector (pSpCas9(BB)-2A-Puro, PX459; Addgene, MA, USA) from the same source as PX458. The recombinant plasmids were transfected into male and female Yorkshire pig fibroblasts. The cleavage efficiency of the recombinant plasmids in the fibroblast clones was detected by T7EN1 (New England Biolabs, Ipswich, MA, USA) analysis as described by [27]. The nuclei of *TRBV*-edited fibroblasts were then transferred into enucleated oocytes collected from female Yorkshire pigs (Yunnan Shennong Agricultural Industry Group Co., Ltd., Kunming, China) via somatic cell nuclear transfer (SCNT) following a protocol described previously [32]. The reconstructed embryos were surgically transferred into the oviducts of surrogate Duroc pigs with a reproductive history. Pregnancy in the recipients was diagnosed on day 23 after embryo transfer using an ultrasound scanner (HS-101 V, Honda Electronics Co., Ltd., Toyohashi, Japan). The cellular materials and ear tissues were collected to extract genomic DNA using a Tissue DNA Kit (TaKaRa, Shiga, Japan). The mutation types were assessed by PCR using designed primers that covered the entire length of the *TRBV27* gene.

### 2.3. Live African Swine Fever Virus

The ASFV isolate Pig/Heilongjiang/2018 (Pig/HLJ/18) used in this study was provided by Professor Dongming Zhao at the State Key Laboratory of Veterinary Biotechnology, Harbin Veterinary Research Institute, Chinese Academy of Agricultural Sciences, Harbin, People’s Republic of China. This is the first reported ASFV in China and is highly virulent in domestic pigs [33]. The virus was collected from primary porcine alveolar macrophages (PAMs) of ASFV-infected pigs at 20 to 30 days of age. The ASFV-infected PAMs were cultured in RPMI-1640 medium (Thermo Scientific, Waltham, MA, USA) supplemented with 10% FBS at 37 °C with 5% CO_2_ until infection.

### 2.4. Detection of African Swine Fever Virus

The sera, oral and anal swabs, and various tissues and organs of dead animals from ASFV-infected pigs were collected on day five. To obtain a similar amount of material, the oral and anal swabs were collected in the same way. Specifically, the marked swabs were inserted into the mouth and anus at the same depth and in the same position and were rotated five times. The amount of blood and tissue samples used for DNA extraction was 100 µL and 10 g, respectively. Viral genomic DNA was extracted from the above samples and positive and negative samples using GenElute™ Mammalian Genomic DNA Miniprep Kits (Sigma Aldrich, St. Louis, MI, USA). qPCR targeting the conserved region of the p72 gene was performed on a QuantStudio 5 system (Applied Biosystems, Waltham, MA, USA) using the same DNA content according to the WOAH-recommended method described previously. The viral load in different samples collected from ASFV-infected pigs was determined by calculating the p72 gene copy number [33,34].

### 2.5. Animal Infection Experiments

Fibroblasts from F0 *TRBV27* edited piglets were used as donors for the production of F1 gene-edited pigs. Eight-month-old *TRBV*-edited and wild-type (WT) pigs in the experimental (n = 4) and control (n = 4) groups were subjected to ASFV infection. The infection experiments were conducted in the enhanced biosafety level 3 (P3+) and level 4 (P4) facilities of the Harbin Veterinary Research Institute (HVRI) of the CAAS and were approved by the Ministry of Agriculture and Rural Affairs. The two groups of experimental pigs were fed in the same room for one week to acclimate before ASFV infection. Prior to infection, whole blood samples were collected from each pig in both groups and analyzed for complete blood count (CBC) using a veterinary hematology analyzer (ProCyte Dx, IDEXX Laboratories, Westbrook, ME, USA). For ASFV infection, all pigs were inoculated with the virus at a dose of 10^2^ HAD50 via intramuscular (IM) injection. The pigs were raised under normal conditions.

### 2.6. Monitoring of Infection Intensity and Disease Process

During the 10-day infection period, the rectal temperature and clinical signs (mental status; appetite; and skin, respiratory, and gastrointestinal signs) of the pigs in both groups were observed daily. Additionally, daily viral monitoring was performed on blood and oral and rectal swabs collected from each ASFV-infected pig. The deceased animals were immediately necropsied, and their heart, liver, spleen, lungs, kidneys, tonsils, and vital lymph nodes (submandibular, inguinal, mesenteric, bronchial, and hepatogastric) were collected for gross lesion examination. As previously described for pathogenicity assessment [35], the gross lesions were classified into four levels: no obvious changes and mild (+), moderate (++/+++), or severe lesions (++++).

### 2.7. Transcriptome Analysis of the TRBV Gene Family

The pig reference genome (*Sus scrofa* 11.0) was downloaded from the National Center for Biotechnology Information (NCBI) GenBank database (https://www.ncbi.nlm.nih.gov/nuccore/NC_010460.4?report=genbank, Bethesda, MD, USA). RNA-seq data were obtained from the spleens of three biological replicates of ASFV-infected and healthy pigs (https://www.ncbi.nlm.nih.gov/bioproject/PRJNA778812; NCBI, Bethesda, MD, USA) [36]. The reads of each sample processed by FastQC v 0.11.7 (quality control) [37], Trimmomatic v 0.38 (adapter and low-quality read trimming) [38], and BBMap 38.16 [39] were mapped to the pig reference genome using STAR v 2.5.3a [40]. Among the 38 members of the pig *TRBV* family reported in a previous study, 26 are expressed and functional, including the *TRBV27* analyzed in this study [23]. To understand their expression levels in ASFV-infected pigs, the results processed by STAR were used to further calculate the reads mapped to each of the 26 *TRBV* genes using SAMtools 1.16 [41]. The number of reads counted for each gene was displayed in a heatmap generated by TBtools v1.112 [42]. The number of reads mapped to the three CDR domains of the 26 *TRBV* genes was visualized using IGV 2.16.2 [43].

### 2.8. Statistical Analysis

The differences in temperature, complete blood count, and viral load between the gene-edited and wild-type pigs were statistically analyzed using *t*-test in the R package stats (https://www.rdocumentation.org/packages/stats/versions/3.6.2; RStudio; Boston, MA, USA). A *p* value less than 0.05 indicated a significant difference.

## 3. Results

### 3.1. Expression Profile of the TRBV Gene Family in ASFV-Infected Pigs

Of the 38 members of the *TRBV* gene family reported in domestic pigs, 26 are commonly expressed and potentially functional, and all possess the CDR1, CDR2, and CDR3 domains. Transcriptome analyses revealed that four of these genes (the *TRBV3*, *TRBV5-3*, *TRBV11*, and *TRBV14* genes) were not expressed in either wild-type or ASFV-infected pigs. In contrast, the expression levels of the remaining 22 *TRBV* genes–particularly the *TRBV4-4*, *TRBV10*, *TRBV24*, and *TRBV25* genes—were higher in ASFV-infected pigs than in wild-type pigs. Among them, the expression levels of the *TRBV27* gene and the remaining *TRBV* genes were higher in ASFV-infected pigs than in wild-type pigs, except for the *TRBV24* gene, which had the highest expression level (Figure 2A). Further analysis revealed that CDR1 was expressed in the *TRBV27* gene and most of the 22 expressed *TRBV* genes (20/22, 90.9%), (Figure 2B). This suggests that the CDR1 encoded by the *TRBV27* gene in domestic pigs may have a unique function compared to the orthologous gene that has lost the ability to encode CDR1 in African suids.

### 3.2. Generation of TRBV27-Edited Pigs

Four sgRNAs were designed based on the Yorkshire pig genome to develop a CRISPR/Cas9 system for simulating *TRBV27* gene mutations in African suids. The PX458 plasmids containing the designed sgRNAs were transfected into PIECs, and only sgRNA 1 and sgRNA 4 had editing efficiency (Appendix A). The recombinant PX459 plasmid was constructed using this sgRNA pair and then transfected into male and female Yorkshire pig fibroblasts (Figure 3A). The editing efficiency was 76.3% (61/80) and 43.5% (20/46) in the male and female colonies, respectively (Figure 3B). Sanger sequencing revealed that male colony Y5-5 and female colony Y6-33 both carried biallelic deletion mutations. The female colony carried mutations of 209 bp and 210 bp, while the male colony carried mutations of 211 bp and 217 bp (Figure 3C). The sequence encoding the CDR1 domain was disrupted in both the female and male clones. These two *TRBV27*-edited colonies were subsequently used as donor cells to develop reconstructed embryos via somatic cell nuclear transfer. The F0 generations of six 83-day-old female fetuses and one female piglet from donor cell colony Y6-33 and the F0 generations of three male piglets from donor cell colony Y5-5 were produced (Appendix A). The genotypes of these fetuses and piglets matched their donor cell colonies (Figure 3C and Appendix A). Subsequently, the gene-edited fibroblasts from F0 male piglets (Y5-5-2) and F0 female fetuses (Y6-33-4) were re-cloned by somatic cell nuclear transfer, and 14 of the F1 male piglets and three of the F1 female piglets were successfully generated (Appendix A). After sexual maturity, four of the F1 gene-edited males and four wild-type males were transferred to HVRI for ASFV challenge at eight months of age (Appendix A).

To investigate fertility, the F1 generations of *TRBV27*-edited male and female pigs with normal weight gain were mated, and 13 offspring were successfully produced. Eight of these offspring were stillborn, and the remainder died shortly after birth (Appendix A).

### 3.3. Replication of ASFV in TRBV27-Edited and Wild-Type Pigs

The number of white blood cells (lymphocytes, neutrophils, and basophils) in the gene-edited pigs was significantly lower than that in the wild-type pigs at one day post-infection (DPI) (Figure 4A). During the first three days after ASFV infection, all the pigs maintained normal body temperatures, except for one wild-type pig whose body temperature was close to 40 °C. The body temperatures of both the gene-edited and wild-type pigs rose above 40 °C after DPI 4. The ASFV-infected pigs from the two groups experienced a peak body temperature of more than 41 °C at DPI 5 (Figure 4B). Therefore, sera and oral and rectal swabs were collected at DPI 5 from both groups to detect ASFV replication using qPCR detection of the p72 gene. The results revealed greater viral loads in the rectal swabs and sera of the *TRBV27*-edited pigs than in those of the wild-type pigs (Figure 4C). The differences in viral loads in internal tissues and organs were further detected. The major organs and lymphoid tissues of the *TRBV27*-edited pigs were higher than those of the wild-type pigs, especially the liver, spleen, and bronchial lymph node (LN4) (Figure 4D).

### 3.4. Differences in Clinical Signs and Organ Lesions

All the ASFV-infected pigs died within 10 days, but the wild-type and *TRBV27*-edited pigs showed different clinical signs. The clinical signs in the wild-type pigs were poor appetite and lack of spirit. However, in addition to the clinical signs exhibited by the wild-type pigs, the *TRBV27*-edited pigs exhibited more clinical signs, including difficulty standing, broken skin on the testicles and hooves, paleness, respiratory distress, and abdominal haemorrhagic spot (Appendix A). The dead pigs from both groups were necropsied to visualize the gross lesions in the internal organs and lymphoid tissues. Both the gene-edited and wild-type pigs showed various lesions in the major organs and lymphoid tissues, with haemorrhage being the predominant lesion (Appendix A). Gross lesions and pathogenic scores showed similar severe lesions in organs and lymphoid tissues between the two groups (Appendix A; Appendix A). However, one edited pig had more severe lesions in the liver, lungs, and lymph nodes than the wild-type pig did (Figure 5).

## 4. Discussion

A previous comparative genomic analysis revealed that sequence variation in the *TRBV27* gene is one of the few major differences between the genomes of African suids and domestic pigs. This difference is characterized by the deletion of a 280-bp fragment between exons 1 and 2 of the *TRBV27* gene in African suids, including a sequence encoding the CDR1 domain involved in T cell antigen recognition [24]. In this study, we simulated this variation in the *TRBV27* gene in domestic pigs using the CRISPR/Cas9 system and investigated its effect on the host response to exogenous pathogens using ASFV as an example. The results of infection experiments showed the differences in ASFV infection between the genetically modified pigs and the wild-type pigs. This was reflected by an increase in both viral load and clinical signs in the former groups. The CDR1 domain encoded by the *TRBV27* gene could be associated with the ASFV infection in domestic pigs.

The exacerbation of ASFV infection in gene-edited domestic pigs supports the importance of the *TRBV27* gene in antigen recognition. In vertebrates, TCR α and TCR β are responsible for the recognition of antigenic peptides and play a central role in the adaptive immune response. In the latter, three hypervariable regions (CDR1, CDR2, and CDR3) encoded by the *TRBV* gene family have been identified as key antigen recognition components [19,20]. The former two are responsible for interacting with the antigen-presenting molecule MHCs [23]. It is therefore reasonable to speculate that deletions or mutations in these two domains may lead to changes in T cell-mediated antigen recognition, resulting in differences in the host immune response. Before ASFV infection, deletion of the sequence encoding CDR1 in the *TRBV27* gene resulted in a significant reduction in the number of whole blood cells involved in infection defense. Consistent with this, the ASFV in domestic pigs was more pathogenic to wild-type pigs, mainly characterized by higher viral loads and more severe clinical signs. These findings suggest that the *TRBV27* mutation could disrupt the homeostasis of the host immune system and therefore reduce resistance to ASFV in domestic pigs. In contrast, mutations in this gene in African suids may have resulted from adaptive evolution and may be harmless, as they have natural resistance to ASFV [24,44].

The *TRBV27* gene could be a peripheral member of the *TRBV* gene family that participates in antigen recognition. Among the *TRBV* genes known to potentially function, 26 have three CDR domains and are thought to be commonly expressed [23]. In the present study, published transcriptomic data from ASFV-infected pigs were reanalyzed [36] and 22 of the 26 *TRBV* genes were found to be expressed. However, the expression level of the *TRBV27* gene was almost unchanged before and after ASFV infection. In contrast, the expression levels of other *TRBV* genes, particularly the *TRBV4-4*, *TRBV4-5*, *TRBV24*, and *TRBV25* genes, were obviously upregulated in ASFV-infected pigs (Figure 2A). This finding suggested that the *TRBV27* gene may not be a core functional member involved in TCR antigen recognition. Therefore, there were no significant differences in organ or tissue lesions between the *TRBV27*-edited pigs and wild-type pigs after ASFV infection.

The potential functional compensatory effects of the *TRBV* gene family could have hindered the observation of the complete effects produced by the *TRBV27* mutation. A multigene family originated from the expansion of a common ancestor during long-term evolution. The homologous copies commonly exhibit a functional compensatory effect by performing functions similar to or identical to those of the original gene [21,45]. For example, at least eight members of the Arabidopsis Pht1 phosphate (Pi) transporter family are expressed in roots and exert synergistic effects in response to low-phosphorus stress environments [46]. Similarly, a recent analysis of the domestic pig genome revealed that different copies of the *TRBV* gene family with high sequence similarity [23] and more than 22 *TRBV* genes, including the *TRBV27* gene, were upregulated in ASFV-infected pigs. Therefore, the potential compensatory effect of the *TRBV* gene family could be linked to the lack of significant damage caused by the mutant *TRBV27* gene. The role of individual copies of the *TRBV* gene family should be investigated in future studies.

The characteristics of ASFV that attack the immune system may not be conducive to understanding the true function of the *TRBV27* gene. APC-processed antigenic peptides are presented to the T-cell receptor via major histocompatibility complexes (MHCs) and subsequently activate the antigen recognition response [17]. Although CDR1 is essential for TCR antigen recognition, it is dependent on antigen presentation in the previous step. However, ASFV predominantly infects host macrophages that act as antigen-presenting cells [11,12]. This may have affected the observation of the function of the *TRBV* gene located in the T-cell receptor. Therefore, future studies should use other pathogens with no apparent damage to APCs to further validate the function of *TRBV* genes in domestic pigs.

## 5. Conclusions

We have artificially replicated in domestic pigs a key evolution driven by natural selection in African suids, namely the absence of the CDR1 domain encoded by the *TRBV27* gene. The effect of this unnatural alteration on the immune response of domestic pigs was further investigated using ASFV infection. The results of this study indicated that disruption of the CDR1 encoded by the *TRBV27* gene could interfere with ASFV infection in pigs. This study provides initial experimental evidence to further understand the relationship between the evolution of the swine immune system and pathogen infection. Future research is needed to investigate the effects and molecular mechanisms of *TRBV27* mutations on the resistance of domestic pigs to invading pathogens that do not directly attack the host immune system.

## Figures and Tables

**Figure 1 microorganisms-12-01113-f001:**
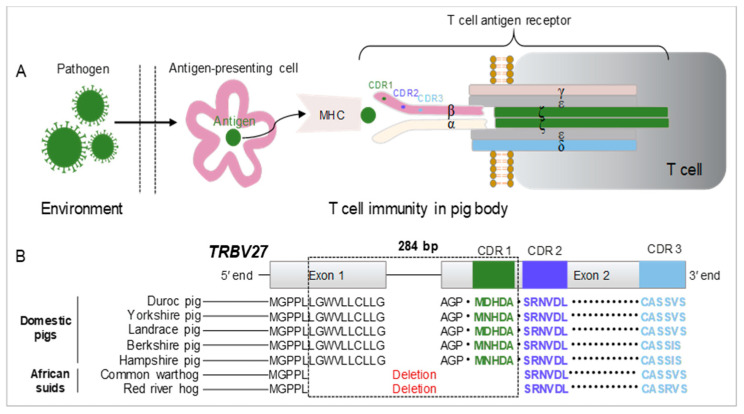
Structural variation in the *TRBV27* gene between the genomes of domestic pigs and African suids. (**A**) Schematic diagram of the swine T-cell immune response to exogenous pathogens. (**B**) Sequence alignment of the *TRBV27* gene among pig genomes. The symbol “·” indicates that the amino acids of the corresponding regions are not shown. Consecutive letters in different colors are the amino acid sequences corresponding to the CDRs immediately above them. MHC: major histocompatibility complex; CDR: complementarity-determining region; α, β, γ, δ, ϵ, and ζ are subunits of the T-cell receptor.

**Figure 2 microorganisms-12-01113-f002:**
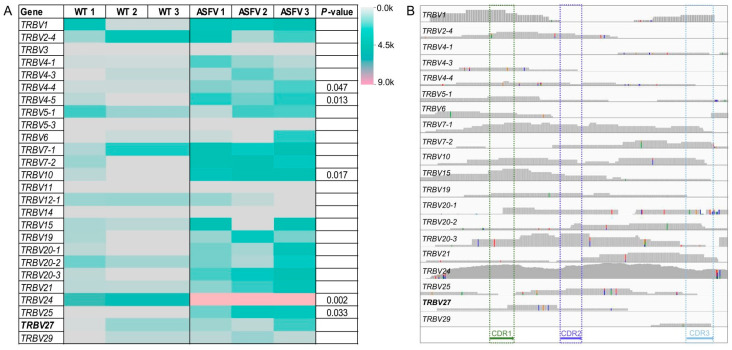
Comparison of *TRBV* gene expression profiles between wild-type (WT) and African swine fever virus (ASFV)-infected pigs. (**A**) A heatmap of the number of RNA-seq reads mapped to the 26 *TRBV* genes with potential functions. k: thousand. (**B**) A bar chart of the number of reads mapped to the three major domains of the 22 expressed *TRBV* genes after ASFV infection. The scale range of the vertical coordinate is 0 to 1000. CDR: complementarity-determining region.

**Figure 3 microorganisms-12-01113-f003:**
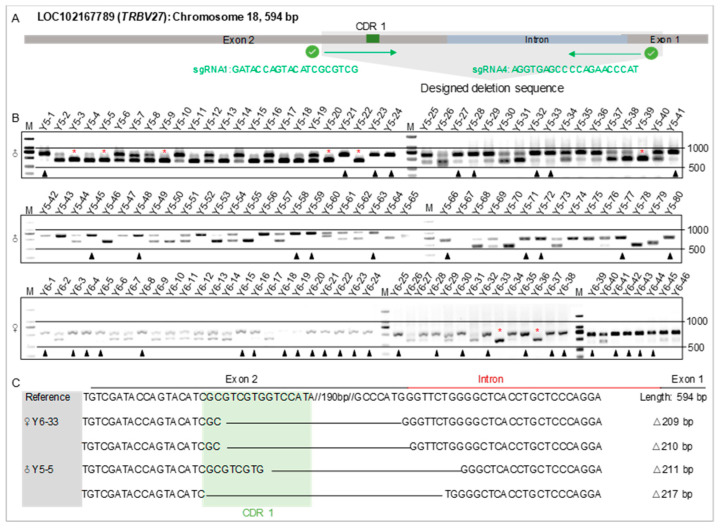
Identification of the editing efficiency of the *TRBV27* gene in porcine fetal fibroblasts. (**A**). Dual sgRNA-directed mutation of the *TRBV27* gene. The gray region shown in the schematic of the *TRBV27* gene is the designed target sequence. (**B**). Verification of sgRNA working efficiency in male and female porcine fetal fibroblasts after digestion with T7 endonuclease I. Black triangles indicate unedited cells, while red asterisks indicate edited cells that did not show the two separate bands. (**C**). Verification of the biallelic gene mutation in *TRBV27*-edited male and female fibroblasts.

**Figure 4 microorganisms-12-01113-f004:**
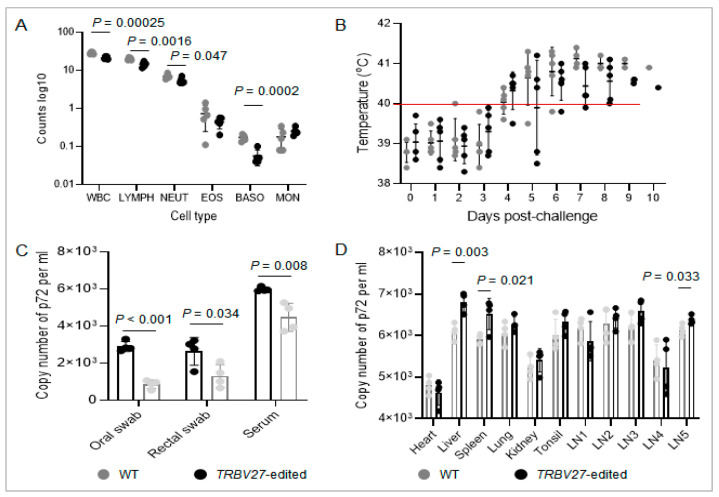
Differences in relevant indicators before and after African swine fever virus infection in *TRBV27*-edited and wild-type pigs. (**A**). Number of immune-related white blood cells detected in the two groups of infected pigs at 0 days post-infection. (**B**). Rectal temperatures of the pigs in the two groups throughout the infection period. (**C**). Viral loads in the body fluids of the two groups of infected pigs at day five post-infection. (**D**). Viral loads in major organs and lymphatic tissues of the two groups of infected pigs at day five post-infection. WBC: white blood cell; LYMPH: lymphocyte; NEUT: neutrophil; EOS: eosinophil; BASO: basophil; Mon: monocyte; LN1: mandibular lymph node; LN2: inguinal lymph node; LN3: mesenteric lymph node; LN4: bronchial lymph node; LN5: hepatoduodenal lymph node; WT: wild type; *p* < 0.05 indicates a significant difference.

**Figure 5 microorganisms-12-01113-f005:**
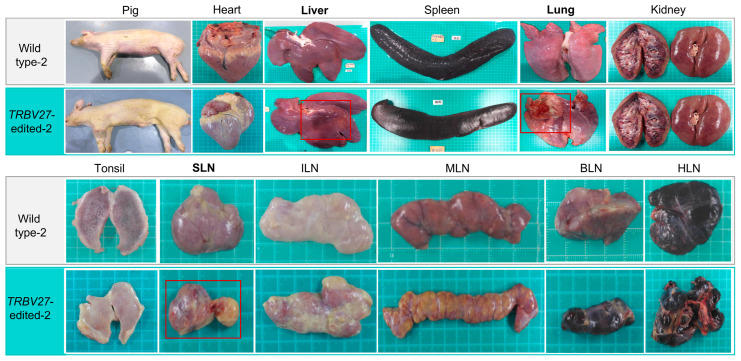
Comparison of organ lesions in one pig from each of the *TRBV27*-edited and wild-type pigs infected with African swine fever virus. Red rectangles indicate more severe damage to the corresponding organ in the gene-edited pigs. Each small white square in the green mat under each organ represents 1 cm. SLN: mandibular lymph node; ILN: inguinal lymph node; MLN: mesenteric lymph node; BLN: bronchial lymph node; HLN: hepatoduodenal lymph node.

## Data Availability

Data are contained within the article.

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
