# Peer review of "Dysfunction of Complementarity Determining Region 1 Encoded by T Cell Receptor Beta Variable Gene Is Potentially Associated with African Swine Fever Virus Infection in Pigs"

_microorganisms, 2024, doi:10.3390/microorganisms12061113_

Round 1
Reviewer 1 Report
Comments and Suggestions for Authors
In the study the Authors investigated the role of TRBV27-encoded CDR1 on the resistance of domestic pigs to ASFV, as a model for an exotic pathogen. The results, obtained through experimental infection of gene-edited and wild type pigs, are different from the hypothesis of a resistance given by the 284bp deletion, yet they can be of interest. The study is correctly designed and the results are well described. The manuscript can be accepted after the following few minor revisions:
Title
- please replace complementary with complementarity
Introduction
- lines 44-45 this sentence is obvious and can be deleted
- line 54: replace OIE with WOAH
Results
The paragraph 3.1 seems to me that it describe what it is aleady known rather than results obtained in the study. It should be moved in the introduction
Discussion
- lines 31: check the english form of the sentence "Therefore, the functioning.....omissis.....on domestic pigs."
Reviewer 2 Report
Comments and Suggestions for Authors
The manuscript by Jiayu Li describes gene editing of domestic pigs to remove a region (284 bp in length) of the beta T-cell receptor V27 gene (TRBV27) that differs between African suids and domestic pigs. The wt and gene edited pigs were then inoculated with African swine fever virus (ASFV) and the course of the infection studied. The authors report that higher viral loads and more severe clinical signs were observed in the gene-edited pigs, that lack this CDR2. However, all pigs did develop severe disease, with a similar time course. Some of the studies are incomplete and there are some ethical concerns as well (see below).
Major points:
- Although the details of the the gene editing are important, they are not really part of the point of this manuscript and the information in Figures 2 and 3 is essentially supplementary information. Indeed the legend for Figure 3 does not really correspond well to the Figure itself. For example for panel E, the legend indicates: “Four F1 male pigs were infected with ASFV” and the image shown is just the 4 pigs, this is totally uninformative. Similarly panels D and G in this Figure just show images of two pigs and nothing is gained from this.
- Was it expected that the WBC counts would be lower in the TRB27 gene-edited pigs (see Figure 4)? Why should this be? Since the PBMCs are major targets for ASFV-infection then do these changes in cell populations make it difficult to interpret the differences in virus yield observed between the wt and edited pigs? In panels C and D it would be good if an extraction control was included in the analysis (using a host gene, e.g. GAPDH) to ensure fair comparisons. There is also a concern about how the data are analysed in panels C and D of Figure 4. The Ct values appear to have been averaged and the means and SD (I assume, it does not appear to be stated) determined as though these are “normal” numbers. In fact Ct values are essentially on a log2 scale, a difference in 10 in Ct value represent 2^10 (=1024) difference in copy number. The assessment of statistical significance should be assessed on normal numbers, after conversion of the Ct values to genome copy numbers. It is not clear why only the data for rectal swabs and serum is shown in Figure 4 panel C. The data for the oral swabs mentioned in the section 2.4 are not shown but should be much more informative than the rectal swabs which only have low levels of viral DNA (high Ct values), see Figure 4C.
- In the animal infection experiment, the text indicates that 8-month old pigs were used (line 144, section 2.5 and line 217), is this correct? Should it be 8-week old pigs? Furthermore, the text on lines 153 and 154 indicates the ASFV-infected pigs were kept until they died. Normally a humane end point is defined in advance of the experiment (for approval by the Animal ethics committee) and the pigs euthanised when this endpoint is reached, to minimise suffering. Sometimes, ASFV-infected pigs can still die suddenly since the progress of the disease can be very quick but such occurrences should be minimised.
- In Figure 5, it is unfortunate that the organs from uninfected pigs are not shown for comparison. This seems an essential “control”.
- In Figure 6, It seems that the TRBV gene expression profiles are for the wt pigs only. It would be good to see the same analysis for the gene edited pigs. Perhaps expression of other members of the TRBV gene family is modified in the gene-edited pigs which may account for some of the differences observed. This is a major omission. It should be apparent that the number of reads mapping to the CDR2 region will be different in the gene edited pigs (see panel B, Figure 6). This analysis is therefore an important and useful control.
Minor points
- TBR27 should be spelled out in the title. I also think the title needs modifying since I do not think “reduced resistance” is demonstrated.
- Line 216-217. The text reads the male pigs” were subjected to HVRI for ASFV challenge”, this needs change since HVRI is the abbreviation used for the Harbin Veterinary Research Institute (see line 147) and thus this does not make sense, maybe “transferred” rather than “subjected”.
- I think the phrase “Therefore , the integrity of the CDR domains encoded by TRBV27 contributes to the defense of domestic pigs against exogenous pathogens” (see Abstract, lines 36-38) cannot be justified on the basis of the experiments described in this manuscript.
In general the English is OK but there are some places where improvement is required including examples given above.
Round 2
Reviewer 2 Report
Comments and Suggestions for Authors
Jiayu Li et al., have revised their manuscript and it is improved in some respects but I still think the authors try to over interpret their results. I am not convinced there is a real difference in susceptibility of the gene-edited pigs compared to the wt pigs. As indicated on lines 277-279, there were no obvious differences in gross lesions and pathogenic scores between the two groups. The transcriptomic analysis (now in Fig. 5) has not been changed and is thus still incomplete. I think the attempt to broaden the significance of the results (see end of Abstract, lines 37-42)) is also unsupported. Just one gene-edited pig (out of 4) had more severe lesion scores than the wt pigs (line 280-281).
The changes in VP72 gene expression should be accompanied by measurements of an extraction control, as mentioned previously, otherwise it is not certain that similar amounts of material (from swab samples) are being examined (now in Figure 4 C)
The revised title still overstates the evidence I think (note Bata should be Beta and it should be “interferes with the infection…”.)
Comments on the Quality of English LanguageOverall it is OK but the revised title need changing as described above.
